# Performance-Based Robotic Training in Individuals with Subacute Stroke: Differences between Responders and Non-Responders

**DOI:** 10.3390/s23094304

**Published:** 2023-04-26

**Authors:** Ophélie Pila, Christophe Duret, Typhaine Koeppel, Pascal Jamin

**Affiliations:** 1Centre de Rééducation Fonctionnelle Les Trois Soleils, Médecine Physique et de Réadaptation, Unité de Neurorééducation, 77310 Boissise-Le-Roi, France; c.duret@les-trois-soleils.fr (C.D.); typhaine.koeppel@gmail.com (T.K.); 2Institut Robert Merle d’Aubigné, Rééducation et Appareillage, 94460 Valenton, France; p.jamin@irma-valenton.fr

**Keywords:** responsiveness, hemiparesis, upper extremity, kinematics, robot-based therapy, training modality

## Abstract

The high variability of upper limb motor recovery with robotic training (RT) in subacute stroke underscores the need to explore differences in responses to RT. We explored differences in baseline characteristics and the RT dose between responders (ΔFugl-Meyer Assessment (FMA) score ≥ 9 points; *n* = 20) and non-responders (*n* = 16) in people with subacute stroke (mean [SD] poststroke time at baseline, 54 (26) days, baseline FMA score, 23 (17) points) who underwent 16 RT sessions combined with conventional therapies. Baseline characteristics were compared between groups. During RT sessions, the actual practice time (%), number of movements performed, and total distance covered (cm) in assisted and unassisted modalities were compared between groups. At baseline, participant characteristics and FMA scores did not differ between groups. During the RT, non-responders increased practice time (+15%; *p* = 0.02), performed more movements (+285; *p* = 0.004), and covered more distance (+4037 cm; *p* < 10^−3^), with no difference between physical modalities. In contrast, responders decreased practice time (−21%; *p* = 0.01) and performed fewer movements (−338; *p* = 0.03) in the assisted modality while performing more movements (+328; *p* < 0.05) and covering a greater distance (+4779 cm; *p* = 0.01) in unassisted modalities. Despite a large amount of motor practice, motor outcomes did not improve in non-responders compared to responders: the difficulty level in RT may have been too low for them. Future studies should combine robot-based parameters to describe the treatment dose, especially in people with severe-to-moderate arm paresis, to optimize the RT and improve the recovery prognosis.

## 1. Introduction

Strokes are the leading cause of long-term disability in adults in Western countries. One-third of affected individuals have a significant residual disability at 6 months [1]. Upper limb paresis, the most common motor impairment, negatively affects daily living activities and quality of life [2,3,4]. 

Restoration of upper limb motor function is, therefore, a critical challenge to help survivors to regain independence. The recovery process after a stroke mostly involves spontaneous re-organization of neural networks. This occurs predominantly within the first 3 months, although it can continue for 6 to 12 months in those with severe impairment [5,6,7]. 

Previous studies demonstrated a proportional relationship between the severity of the initial impairment and the magnitude of the upper limb motor recovery [8,9,10]. The recovery of most individuals follows a proportional process that has been formalized as the “70% rule” (i.e., individuals recover 70% to 80% of the difference between their initial upper extremity Fugl–Meyer Assessment (FMA) score and the maximum score (66 points)). However, some studies found that recovery after a severe impairment is highly variable and does not fit this model [11]. This predictive model, described by some authors as over-optimizing [12], seems to be unaffected by the therapy dose [13] and, therefore, questions the effectiveness of current doses of the therapy [14].

Over the last 2 decades, neuroscientific advances [15,16,17,18,19] and clinical trials [20,21,22,23,24,25] have identified training factors that impact activity-induced brain plasticity [26]. A growing body of literature shows that an intensive protocol involving repeated, task-oriented, active movements can improve motor outcomes. Thus, post-stroke rehabilitation management has been improved by modulating qualitative and quantitative treatment parameters, such as the treatment dose and intensity.

Rehabilitation robots can meet these requirements because they enable the delivery of highly repetitive training of task-specific exercises of progressive difficulty in an interactive environment. Clinical results from robotic studies, mainly measured with the FMA, have shown that robotic therapy (RT) reduces upper limb motor impairment to a greater extent than conventional therapy [27]. Furthermore, some studies found that improvements after RT were greater in individuals with severe impairment than less severe impairment [28].

Although improvement in motor function after robot-based upper limb rehabilitation is unequivocal, motor outcomes are the result of individual responses, ranging from no change to a large improvement in motor function. Although this inter-subject variability in training responses cannot yet be fully explained, some studies using neuroimaging or neurophysiology have found that biomarkers such as the functional preservation of the corticospinal tract can predict motor outcomes [29].

This study aimed to explore how individuals who respond to an intensive robot-mediated rehabilitation program (with an improvement of at least the minimal clinically important difference in upper limb motor function: responders differ from those who do not (non-responders) in terms of clinical and kinematic outcomes and treatment dose-related parameters). We hypothesized that RT applied with accurate knowledge of the training dose could improve the practice’s effectiveness and training responses.

## 2. Materials and Methods

### 2.1. Study Design and Sample

This retrospective study used data collected from standard care in the rehabilitation department of the Centre de Réadaptation Fonctionnelle (CRF) Les Trois Soleils (Boissise-le-Roi, France) between 2009 and 2019. The study is reported according to the Strengthening the Reporting of Observational Studies in Epidemiology (STROBE) statement. The study is ancillary to a previously published study describing the dose of physical treatment administered to patients during rehabilitation sessions using a robotic device [30]. The study was performed in accordance with current French legislation (reference N° 004 (MR004)) and was approved by our internal ethics committee in line with the data protection act. It was registered on the Health Data Hub (N° F20211206141427).

We used data from 36 individuals with subacute stroke who completed an upper limb rehabilitation program of RT combined with occupational therapy in usual care. Participants were divided into two groups, responders and non-responders, according to the magnitude of response to the program defined by the change in FMA score. Participants were identified as responders if they had a change from baseline to post-program of ≥9 points on the FMA. This 9-point change was previously determined as the minimal clinically important difference (MCID) in individuals in the subacute phase of stroke [31].

### 2.2. Interventions

Participants completed a 4 week upper limb program of RT and occupational therapy 3 to 4 times per week. Conventional occupational therapy sessions included mobilization and stretching of the paretic upper limb to improve or maintain joint range of motion, individual joint and whole upper limb exercises, fine motor control, and grasping exercises to improve sensation and functional and fun exercises involving both motor and cognitive functions. RT was delivered using the InMotion Arm^®^ robot (InMotion 2, Interactive Motion Technologies, Inc., Watertown, MA, USA), a 2-degree-of-freedom distal effector-type manipulandum that trains shoulder and elbow movement in the horizontal plane. 

Participants moved the manipulandum (Figure 1) to perform a circular pointing task that involved repeated center–out movements (and back to center) towards 8 visual targets located around a circle with a 14 cm radius. Participants performed movements either with or without robotic assistance using the assisted or unassisted modality. The level of assistance was modulated by the system in accordance with the participant’s performance. Four center–out distances to achieve were available (3, 5, 10, and 14 cm), and the distance was selected by the therapist according to the person’s motor ability.

### 2.3. Data Measurement

Demographic and other data, such as age, sex, side of paresis, type of stroke, and time since stroke at RT initiation, were collected. Upper limb impairment was assessed pre- and post-program using the FMA. This evaluation is reliable and sensitive to change and has been validated for use in individuals with spastic paresis in the subacute phase of stroke. The difference between the FMA before and after the combined program was used to categorize the participants as responders and non-responders. Participants with subacute stroke who had an improvement in FMA ≥ 9 points were classified as responders. 

The robot-based evaluation was performed at pre- and post-program, and kinematic raw data were extracted. This evaluation consisted of a pointing task that involved 80 non-assisted center–out and out–center movements to 8 targets located around a 14 cm radius circle before and after the intervention. From the collected kinematic data, 4 kinematic variables were computed:-velocity: defined as the distance traveled divided by the movement time (in cm/s);-distance: defined as the distance between the center of the pointing task and the orthogonal projection on the axis (center-target) of the position of the end-effector at the end of the movement (in cm);-smoothness: defined as the number of peaks in the velocity profile;-accuracy: defined as the root mean square error from the straight line (in cm).

Robot-derived kinematic variables were then normalized to control data (as %) for 3 of the 8 directions [32]. A cohort of 40 healthy subjects was used to define normal values for each kinematic variable [33]. Finally, for each exercise performed during the RT sessions, actual practice time (in percentage; normalized to a session length of 60 min), number of movements performed, and center–out distance were collected. For these measurements, the physical training modality was also identified (assisted/unassisted). Furthermore, the total distance covered (in cm) was calculated for each RT session as the sum of the product of the center–out distance traveled by the number of movements performed for each exercise.

### 2.4. Statistical Analysis

Baseline characteristics (age, sex, side of paresis, type of stroke, time since stroke at RT initiation, baseline FMA score, and baseline robot-derived kinematic variables) were compared between responders and non-responders using 2 sample *t*-tests or Fisher’s exact text (for categorical data). Repeated measures analysis of variance (ANOVA) was carried out to analyze pre/post changes in the 4 robot-based kinematic variables between responders and non-responders. Session (session 1 to session 15) * modality (assisted, unassisted) * group (responders and non-responders) interactions were analyzed for 3 robot-based variables (number of movements performed, actual practice time, and distance covered) with the 3-factor ANOVA. A Bonferroni correction was applied for pair-wise comparisons. Significance was set to *p* < 0.05, and SPSS 17.0 was used for all analyses.

## 3. Results

### 3.1. Baseline Characteristics

Twenty participants were identified as responders and 16 as non-responders. Their baseline characteristics are presented in Table 1. At baseline, only the type of stroke tended to be different between groups (responders vs non-responders, *p* = 0.07); in the responder group, 45% had a hemorrhagic stroke and 55% an ischemic stroke, whereas in the non-responder group, 87% had an ischemic stroke and 13% a hemorrhagic stroke.

### 3.2. Clinical Outcomes

Clinical results are reported in Figure 2. The pre/post change in the FMA score (*p* < 10^−4^) differed between groups. At post-program, the FMA score had increased significantly only for responders by +17 (9) pts (mean (SD)).

### 3.3. Kinematic Outcomes

Kinematic results are reported in Figure 2. The pre/post change in distance (*p* = 0.049), smoothness (*p* = 0.01), and accuracy (*p* = 0.004) differed between groups. At post-program, only the distance increased significantly for responders by +49 (47)% (*p* < 10^−3^).

### 3.4. Robot-Based Outcomes with All Physical Modalities Pooled

The results are summarized in Figure 3 and Table 2. There was a between-group difference in the actual practice time (*p* < 10^−5^), number of repeated movements (*p* < 10^−3^), and total distance traveled (*p* < 10^−3^).

During the RT sessions, the actual practice time decreased by a mean of 15 (16)% for responders (S2 vs. S15, *p* = 0.003). At the same time, the actual practice time, number of repeated movements, and total distance traveled increased for non-responders (all significant comparisons are shown in Table 2).

### 3.5. Robot-Based Outcomes for Each Physical Modality

The results are presented in Figure 4. Significant interactions between the session and modality and the group were found for the actual practice time (*p* = 0.02), number of repeated movements (*p* = 0.01), and total distance traveled (*p* < 10^−3^).

During the RT sessions, actual practice time in the assisted modality decreased for responders by 21 (21)% (session 1 vs. session 15, *p* = 0.01). The number of repeated movements in the assisted modality decreased for responders between session 2 and session 14 by 338 (370) movements (*p* = 0.03) and between session 2 and session 15 by 355 (376) movements (*p* = 0.01); it increased in the unassisted modality between session 1 and session 15 by 290 (299) movements (*p* = 0.049). At session 14, the total distance traveled in the unassisted modality increased for responders by 4779 (5094) cm (vs. S1, *p* = 0.01) and by 4607 (5223) cm (vs. S2, *p* = 0.02).

## 4. Discussion

The purpose of this retrospective study was both to explore upper limb clinical and kinematic profiles before an intensive combined (usual care and RT) rehabilitation program and to compare the dose of RT between participants who achieved the MCID and those who did not. Despite differences in upper limb motor recovery after the intensive program, the level of initial motor impairment was similar between responders and non-responders. Furthermore, the modalities of the transition from using the assisted modality to the unassisted RT modality differed significantly between responders and non-responders in RT, highlighting differences in the level of difficulty of the administered treatment between groups.

### 4.1. Factors Influencing Favorable Motor Outcomes

The retroactive design highlighted the fact that both groups had similar FMA scores at the start of the intensive combined program. Both groups included participants with severe-to-moderate upper limb motor impairment. The prognosis for these individuals is usually unfavorable since a high level of impairment in the early subacute phase is generally associated with poor upper limb recovery [34]. However, the results of this study showed that 56% of the participants experienced improvements that exceeded the MCID. This magnitude of improvement in this time window (between 1.8- and 2.9-months post-stroke) was relatively high compared to typical spontaneous improvement. The 17-point increase in FMA score that occurred in responders is much greater than the 5-point improvement with spontaneous recovery found in a previous longitudinal study [35]. If we consider the type of stroke, the non-responder group included mainly ischemic strokes, which could be considered a limiting factor for recovery. However, given the advances in the management of ischemic strokes at the acute phase, this is no longer the case [36]. Studies of outcome prediction have shown that various factors can explain post-stroke motor recovery [37]. Some studies reported that the initial level of motor deficit was not reliable enough to predict motor outcomes in people with severe impairment because of the high degree of variability [8,10]. Neuroanatomical and neurophysiological features, such as corticospinal tract integrity and the presence of motor-evoked potentials, may have contributed to the positive response to the treatment in the present study. A previous study found that people with smaller baseline motor-evoked potentials benefited the most from RT of the paretic arm [38]. Furthermore, another study showed that individuals in the chronic phase of stroke did not respond to robotic hand movement training if they lacked corticospinal integrity [39]. The degree of the corticospinal tract injury is a significant predictor of motor recovery in individuals with initial severe motor deficits (i.e., FMA < 35 points), but it is not a better predictor than the initial FMA score [29]. As previously suggested, the method of assessing a corticospinal tract injury and the prediction model need to be enhanced to refine the prediction of motor recovery [40]. In addition, the intensity threshold to activate endogenous recovery processes may also explain the smaller degree of recovery in the non-responders, as demonstrated in animal models [41,42]. Rehabilitation intensity is an important factor to trigger significant reorganization [43], but different methods can be used to assess treatment intensity and/or dose.

### 4.2. Toward a Meaningful and Multidimensional Description of Administered Treatment Dose

After a stroke, the dose of treatment, which many studies have quantified, provided to individuals in their rehabilitation program by the time scheduled for therapy and/or time spent in intervention led to the observation of a positive relationship between this parameter and motor outcomes after therapy [21]. However, Lohse et al. (2014) suggested that the active time spent in therapy or the number of repeated movements (number of reps/session or per minute) were more meaningful measures of therapy dose. In the present study, the use of robotics allowed for a precise calculation of the number of movements performed per robotic session, and both groups performed more than 700 movements per session. Despite this high number of repeated movements, the improvement did not reach the MCID at the end of the combined program in the non-responders. This result was not consistent with findings from many meta-analyses that suggested that a higher dose of motor interventions was associated with better recovery [21,44]. Consequently, knowledge of the number of repeated movements alone does not provide an accurate insight into the treatment dose. As a complex but crucial issue, the treatment dose was recently refined using a standardized approach by describing the different and constitutive dimensions of the dose in non-pharmacological interventions [45]. Hayward et al. (2021) deconstructed the concept of the dose, which is multidimensional, with systemic links between the different dimensions. In the proposed framework, 3 dimensions are considered to define the internal aspects of the dose in non-pharmacological interventions: task duration, task difficulty, and task intensity. In a well-defined motor task, the duration is defined as the time spent when performing that task, difficulty refers to the intrinsic degree of difficulty of the task, and intensity quantifies the number of times the task is performed. Although it is difficult to objectively measure these variables within conventional treatments, it is possible with some robotic devices; in this study, these variables were automatically computed, and they provided an accurate indication of the characteristics of the internal dimensions. Furthermore, we included an additional variable, the mean total distance traveled by the hand (not considered in Hayward’s framework), that seemed of interest since it is specific to the pointing task performed during the RT. Combining this distance with the number of movement repetitions allows a more precise quantification of the intensity achieved during the task [30].

### 4.3. Repeated Movements and Difficulty: Synergy and Misunderstanding?

A question raised by the present study is why a differential pattern of motor recovery was observed despite the high number of repeated movements performed by both groups (>700 movements per session). A previous robotic study [46] showed that the RT group (>900 assisted movements per session) improved by approximately 8.7 points on the FMA after 30 sessions over 6 weeks, whereas the control group (without RT) improved by approximately 3.6 points. In the present study, in which the initial level of the motor deficit was similar between both groups and to the participants in the study by Sale et al., the magnitude of improvement in the non-responder group was similar to that of the control group in the study by Sale et al. (without RT) and that of the responders was twice that of the robotic group in Sale et al. Despite strong evidence showing that highly repetitive practice is an active ingredient of rehabilitation that leads to significant motor recovery [1,43], the present finding showed that other factors potentially promote brain plasticity, an issue already raised by Khan et al. [47]. In terms of the RT dose, the results showed that non-responders had a higher actual practice time, performed a greater number of movements, and traveled further distances during RT, whereas no change was found in these 3 variables for responders. However, responders used the physical assistance modality differently: the actual practice time and the number of movements decreased in the assisted modality while the total distance traveled increased in the unassisted modality. The level of difficulty in performing a movement has been demonstrated to promote brain plasticity [22,23,48] and was modulated in the RT by a remarkable transition profile from the use of the assistance to the non-assistance modality. This finding supports the idea that a high number of movement repetitions alone is not sufficient to induce motor recovery [49]. Beyond the number of repeated movements, activity-induced plasticity is related to the degree of difficulty in the motor task [50]. Several studies demonstrated long-lasting changes in cortical excitability following skilled motor training, whereas the repetition of a non-skilled motor task or passive training usually results in no, or only minor, excitability changes. Although many studies have focused on movement repetition rather than the effort associated with movement, it would appear that both factors should be combined to promote motor recovery. A subtle balance between these 2 factors must be found depending on the initial severity of the motor deficit and fatigability.

### 4.4. Robotic Assistance for Individuals with Severe Motor Impairment: Facilitator or Deleterious to Motor Recovery?

The impact of physical assistance on movement practice in RT is still debated despite the large number of studies that demonstrate additional therapeutic benefits of assistance on motor function. In RT, mechanical assistance is mostly based on the individual’s performance, and it largely increases the number of movements per session (>500 movements) as compared with conventional therapy. In the present study, however, we found that non-responders who performed more than 700 movements failed to reach the MCID. At the same time, those individuals required the most movement assistance during the RT sessions. As demonstrated in the lower limb, assistance could minimize effort intensity and cortical activation [51]. In the upper limb, a pilot study that compared assisted (use of assistance) RT with unassisted reaching therapy suggested that movement smoothness improved more when movements were performed without assistance [52]. This result is consistent with that of our study since smoothness improved in the responders who transitioned from the assisted modality to the unassisted modality in RT. A similar improvement occurred in the accuracy parameter. Regarding the distance traveled by the participant’s hand, statistically significant improvements at the end of the intensive program were only found for responders. The present findings question those of a previous study that showed that robotic assistance improved voluntary movement performance in individuals with a severe-to-moderate stroke [53]. Although the assistance delivered in RT was optimized according to motor performance [54,55], we can hypothesize that the non-responders adopted a slacking behavior since they could not take over the assistance provided. They may have needed more time/practice to benefit from the assistance and for motor improvements to occur.

### 4.5. Limitations

This study has several limitations. First, the retrospective design means that the results may be subject to selection, implementation, and evaluation bias. As this study was not randomized or controlled, the sampling method (convenience sample) did not prevent a potential selection bias that can lead to a lack of representativeness of the study sample in relation to the target population. Moreover, it would be interesting to standardize the administration of RT by controlling the difficulty of the task in terms of the physical modalities used: it is important that the level of difficulty proposed is adjusted to the individual (neither too easy, nor too difficult). Finally, we used an indirect method to reflect a change in arm function. Future studies should include a functional assessment. The lack of data on disorders associated with the motor deficit, such as aphasia and visuo-spatial neglect, prevents a global vision of the interpretation of the results. In addition, the fact that clinical evaluations were not performed at regular time points during the combined program limits the interpretation of the transition from assisted to unassisted movement. The lack of neuroimaging data strongly limits the interpretation of the results; future studies should seek to determine if responders have an intact corticospinal tract and non-responders do not.

## 5. Conclusions

This study showed that motor recovery does not depend solely on the initial level of upper limb motor impairment in individuals with a subacute stroke. The difficulty of the movement performed seems to be a determining factor in motor recovery after moderate-to-severe upper extremity paresis. The physical modalities (assisted/unassisted) of RT must be varied to continuously challenge individuals. The timing of the transition from active-assisted to non-assisted modality depends on the individual’s performance during training but seems to be a key to the difficulty of training. Using a combination of robot-based parameters to describe the treatment dose can guide the therapist to tailor the treatment so that individuals achieve clinically significant improvements in upper extremity function. This work can assist clinicians and engineers in the design of future robotic devices to adapt their algorithms and in the design of future clinical studies.

## Figures and Tables

**Figure 1 sensors-23-04304-f001:**
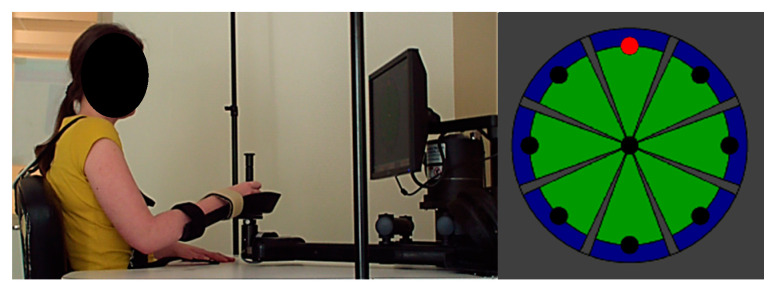
InMotion 2.0 shoulder/elbow robotic system and pointing task interface used.

**Figure 2 sensors-23-04304-f002:**
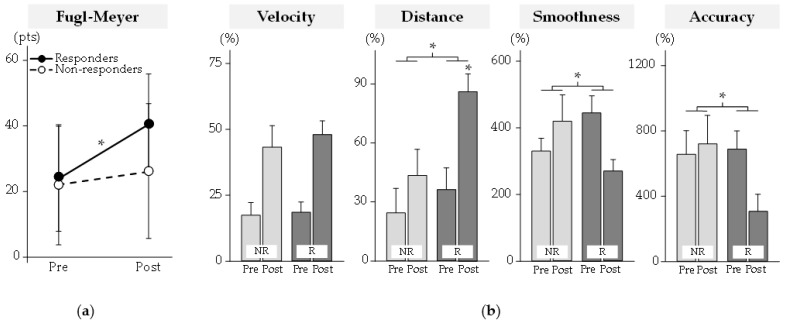
Clinical and robot-based outcomes. (**a**) Clinical outcomes. (**b**) Robot-based kinematic outcomes. Kinematic variables were normalized to control data (as %). NR: Non-Responders; R: Responders. * Significant difference at *p* < 0.05.

**Figure 3 sensors-23-04304-f003:**
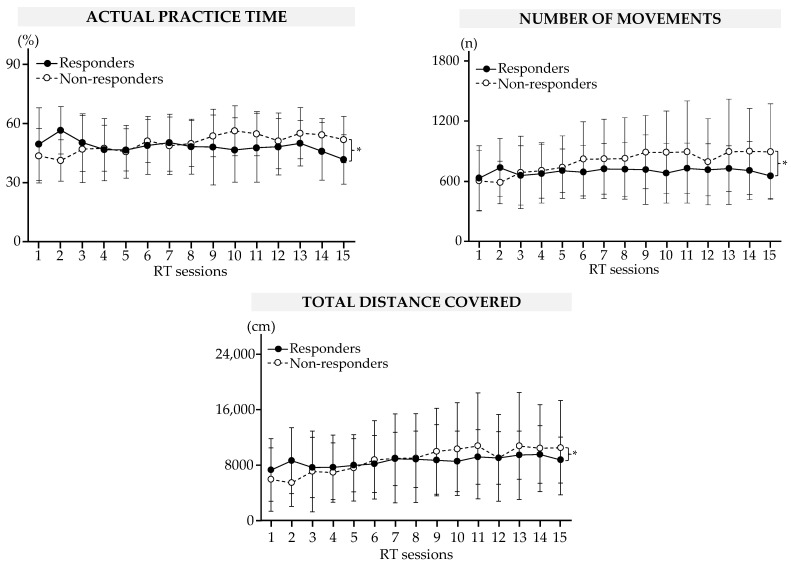
Results of the robot-based variables during RT sessions in responders and non-responders. Data are mean (SEM). * Significant difference at *p* < 0.05.

**Figure 4 sensors-23-04304-f004:**
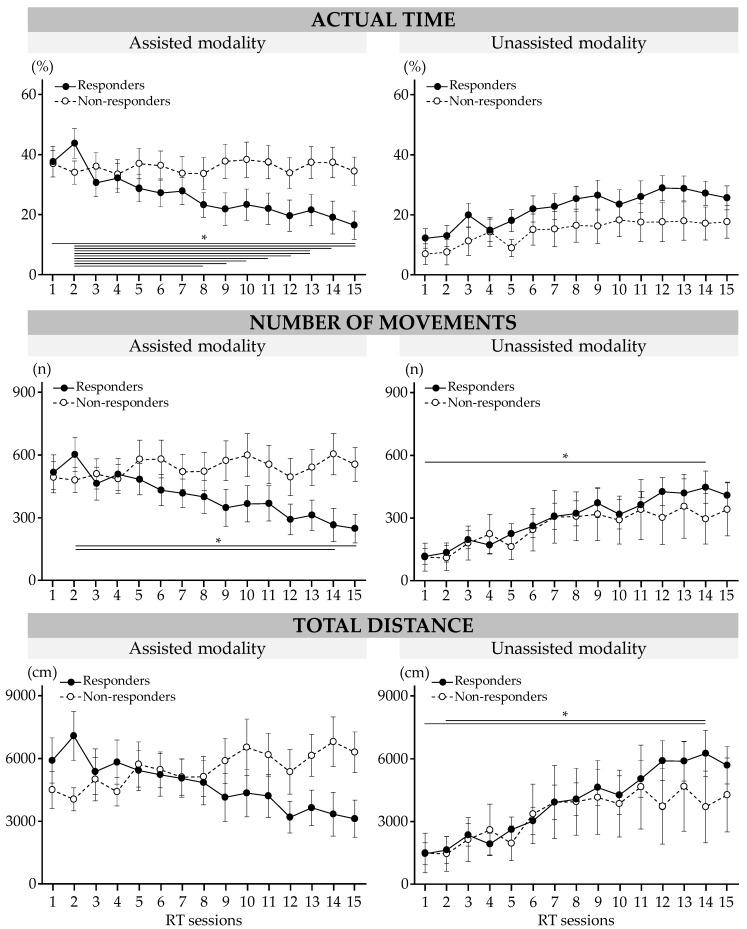
Results of robot-based variables during RT sessions in responders and non-responders. Data are mean (SEM). * Significant difference at *p* < 0.05.

**Table 1 sensors-23-04304-t001:** Baseline characteristics of participants.

	Non-Responders (*n* = 16)	Responders (*n* = 20)
Age (years)	62 (16)	56 (16)
Sex (*n* females/*n* males)	7/9	8/12
Side of paresis (*n* right/*n* left)	10/6	9/11
Type of stroke (*n* ischemia/*n* hemorrhage)	14/2	11/9
Time since stroke at RT initiation (days)	63 (31)	48 (20)
FMA score (66 pts)	22 (18)	24 (16)
Velocity (%)	19 (17)	18 (20)
Distance (%)	37 (49)	25 (50)
Smoothness (%)	447 (230)	329 (158)
Accuracy (%)	689 (497)	657 (580)

Data are the mean (SD). R: right; FMA: Fugl–Meyer Assessment.

**Table 2 sensors-23-04304-t002:** Results of the robot-based variables during RT sessions in responders and non-responders.

	Actual Practice Time (%)		Number of Movements (*n*)		Total Distance Covered (cm)	
	NR	R	NR	R	NR	R
S1	44 (14)	50 (18)	609 (299)	636 (322)	6004 (4540)	7375 (4506)
S2	42 (10)	57 (12)	592 (211)	740 (288)	5521 (3389)	8713 (4737)
S3	47 (17)	51 (15)	692 (360)	662 (294)	7155 (5788)	7737 (4357)
S4	48 (12)	47 (16)	711 (276)	681 (292)	7017 (4280)	7746 (4641)
S5	46 (13)	47 (11)	745 (310)	710 (216)	7683 (4778)	8051 (3817)
S6	51 (11)	49 (15)	826 (369)	695 (265)	8827 (5624) ^b^	8254 (4107)
S7	49 (15)	51 (14)	827 (395)	728 (251)	9043 (6388) ^b^	8977 (3830)
S8	50 (12)	49 (14)	831 (405)	724 (270)	10,041 (6165) ^b^	8921 (4057)
S9	54 (11)	48 (19)	893 (365) ^ab^	720 (345)	10,385 (6673) ^ab^	8776 (5108)
S10	57 (13) ^b^	47 (16)	892 (411) ^ab^	685 (296)	10,830 (7613) ^abd^	8617 (4358)
S11	55 (11)	48 (17)	897 (508) ^ab^	732 (250)	9094 (6222) ^abcd^	9256 (3929)
S12	51 (14)	48 (14)	798 (429)	719 (259)	10,819 (7683) ^b^	9098 (3795)
S13	55 (13)	50 (12)	898 (524) ^ab^	732 (227)	10,499 (6242) ^abcd^	9532 (3461)
S14	54 (8)	46 (15)	902 (426) ^ab^	711 (289)	10,104 (5677) ^abcd^	9599 (4148)
S15	52 (12)	42 (12) ^b^	898 (477) ^ab^	657 (222)	10,567 (6785) ^abcd^	8814 (3301)

Data are mean (SD). Actual practice time was normalized to a session length of 60 min. NR: non-responders; R: responders. a vs. S1: *p* < 0.05; b vs. S2: *p*< 0.05; c vs. S3: *p* < 0.05, d vs. S4: *p* < 0.05.

## Data Availability

All data are available in electronic format at the Centre de Réadaptation Fonctionnelle (CRF) Les Trois Soleils.

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
