# Peer review of "Performance-Based Robotic Training in Individuals with Subacute Stroke: Differences between Responders and Non-Responders"

_sensors, 2023, doi:10.3390/s23094304_

Round 1

Reviewer 1 Report

The article presents a comparison of baseline characteristic differences between responders and non-responders to different doses of robotic therapy for upper limb rehabilitation. The study involved 36 patients, of whom 20 were responders and 16 were non-responders, and the experiments were conducted mainly within the first 3 months after the stroke. The main kinematic baseline characteristics were indirectly determined through robot measures, including velocity, travel distance, smoothness, and accuracy. The data collected in experiments conducted with 40 healthy subjects were considered as normal reference values.

Cnverning english and write style, the article is well written and well-organized, and the English is excellent. This reviewer has no improvement suggestions, despite not being a native speaker. However, a typo was found in line 188.

The Materials and Methods section is clear and objective. In terms of the study population, 36 individuals with subacute stroke were included in the experiments, and after undergoing robotic therapy combined with usual care, they were categorized into responders and non-responders based on whether their change in the Fugl-Meyer Assessment was equal to or greater than 9 after the intervention. It appears that the authors have their own definition of responder and non-responder, which is only clarified in section 2.1. It would be helpful for the reader's understanding if the definition were introduced earlier in the text.

The Discussion section is the most significant contribution of this work as it compares the findings with the current state-of-the-art research. The authors have made a detailed comparison of their results with other studies in the field. However, a drawback of this work is present in Subsection 4.5. I consider the drawbacks valuable as it opens up possibilities for further research by peers. In my opinion, it would be useful to explore the limitations related to selection, implementation, and evaluation biases and suggest ways to overcome them, if possible.

Finally, the abstract needs to be reviewed as it contains many acronyms, which can make it difficult to understand for readers who are not familiar with statistics. To improve it, the authors could use more descriptive language and fewer numbers, making it more accessible to a wider audience. It would be interesting to present a summary of the valuable discussions, emphasizing the most significant ones, as presented in the conclusion paragraph.

Author Response

Response to Reviewer 1 Comments

The article presents a comparison of baseline characteristic differences between responders and non-responders to different doses of robotic therapy for upper limb rehabilitation. The study involved 36 patients, of whom 20 were responders and 16 were non-responders, and the experiments were conducted mainly within the first 3 months after the stroke. The main kinematic baseline characteristics were indirectly determined through robot measures, including velocity, travel distance, smoothness, and accuracy. The data collected in experiments conducted with 40 healthy subjects were considered as normal reference values.

Point 1: Cnverning english and write style, the article is well written and well-organized, and the English is excellent. This reviewer has no improvement suggestions, despite not being a native speaker. However, a typo was found in line 188.

Response 1: We thank the reviewer, the typo has been corrected.

Point 2: The Materials and Methods section is clear and objective. In terms of the study population, 36 individuals with subacute stroke were included in the experiments, and after undergoing robotic therapy combined with usual care, they were categorized into responders and non-responders based on whether their change in the Fugl-Meyer Assessment was equal to or greater than 9 after the intervention. It appears that the authors have their own definition of responder and non-responder, which is only clarified in section 2.1. It would be helpful for the reader's understanding if the definition were introduced earlier in the text.

Response 2: The minimal clinically important difference (MCID) for the Fugl-Meyer assessment scale has previously been used to categorize participants as responders and non-responders in studies addressing the prediction of upper extremity motor recovery in the chronic and subacute phases of stroke [1,2]. The concept of responders and non-responders has been added to the objective of this work at the end of the introduction.

[1] Hamaguchi T, Yamada N, Hada T, Abo M. Prediction of Motor Recovery in the Upper Extremity for Repetitive Transcranial Magnetic Stimulation and Occupational Therapy Goal Setting in Patients With Chronic Stroke: A Retrospective Analysis of Prospectively Collected Data. Front Neurol. 2020 Oct 20;11:581186. doi: 10.3389/fneur.2020.581186.

[2] Lee JJ, Shin JH. Predicting Clinically Significant Improvement After Robot-Assisted Upper Limb Rehabilitation in Subacute and Chronic Stroke. Front Neurol. 2021 Jul 1;12:668923. doi: 10.3389/fneur.2021.668923

Point 3: The Discussion section is the most significant contribution of this work as it compares the findings with the current state-of-the-art research. The authors have made a detailed comparison of their results with other studies in the field. However, a drawback of this work is present in Subsection 4.5. I consider the drawbacks valuable as it opens up possibilities for further research by peers. In my opinion, it would be useful to explore the limitations related to selection, implementation, and evaluation biases and suggest ways to overcome them, if possible.

Response 3:  A paragraph has been added in discussion section, subsection 4.5: “As this study was not randomized or controlled, the sampling method (convenience sample) did not prevent a potential selection bias that can lead to a lack of representativeness of the study sample in relation to the target population. Moreover, it would be interesting to standardize the administration of RT by controlling the difficulty of the task in terms of the physical modalities used: it is important that the level of difficulty proposed is adjusted to the individual (neither too easy, nor too difficult). Finally, we used an indirect method to reflect a change in arm function. Future studies should include a functional assessment.”

Point 4: Finally, the abstract needs to be reviewed as it contains many acronyms, which can make it difficult to understand for readers who are not familiar with statistics. To improve it, the authors could use more descriptive language and fewer numbers, making it more accessible to a wider audience. It would be interesting to present a summary of the valuable discussions, emphasizing the most significant ones, as presented in the conclusion paragraph.

Response 4:  Thank you for pointing this out, the description of the results in the abstract has now been completely changed and important elements of the discussion have been included at the end of the abstract.

Reviewer 2 Report

This study aims to detect the relationship of clinical, kinematic and dose-related parameters of responsiveness to robotic-rehabiliation program. However, the contributions of the work is unclear. Meanwhile, the article writing is far from satisfactory, which are listed as follows:

1. There is no "related works" to confirm the question mentioned in Introduction. For example, Paragraph 4, Line 50, "Over the last 2 decades, ...", the author should list related references of related neuroscientific advances and clinical trials;

2. Beacause the experiment is based on subacute stroke individuals, ethical concern must be provided;

3. A figure of subjects wearing exoskeleton during data measurement must be added in Section 2.3;

4. Section 2.4, line 135, "Baseline characteristics (age, ...", the right bracket is missing, which may confuse readers;

5. There is no enough description and summarize on results, and the results description is messy. For example, Section 3.4, line 175, the author should emphasize the key points of relevant results instead of listing like a experiment report;

6. There are many abbreviations in Abstract but seem not be used in text part;

7. The innovation and contribution of this article is not clear.

Based on the above mentioned points, I'm sorry to tell you that my advise on this paper is "Reject".

Author Response

Response to Reviewer 2 Comments

This study aims to detect the relationship of clinical, kinematic and dose-related parameters of responsiveness to robotic-rehabiliation program. However, the contributions of the work is unclear. Meanwhile, the article writing is far from satisfactory, which are listed as follows:

Point 1. There is no "related works" to confirm the question mentioned in Introduction. For example, Paragraph 4, Line 50, "Over the last 2 decades, ...", the author should list related references of related neuroscientific advances and clinical trials;

Response 1: Thank you for this suggestion. References have been added.

Point 2. Because the experiment is based on subacute stroke individuals, ethical concern must be provided;

Response 2: Our clinical research is not a prospective study involving human subjects. In France, human research is covered by the “Jardé law”, since 2012 and in accordance with the Declaration of Helsinki, which distinguishes 3 types of human research:

Type 1: interventional “At-risk”

Type 2: interventional “Low-risk”

Type 3: non-interventional

Our study is not covered by the Jardé law because it is retrospective. For this type of study, French regulations only require that the protection of personal data is ensured; ethics approval is therefore not required for our study. However, this study was approved by our internal ethics committee (not declared as an institutional review board) in line with the data protection act. The reference methodologies set out by the Data Protection Commission provide a framework for processing information including health data. Reference methodology No. 004 (MR004) affects research not involving human subjects (i.e., studies and evaluations based on data). In addition, all participants were informed of the use of their data for research purpose and their consent was recorded in the medical records. Furthermore, the data were pseudonymized to guarantee the protection of personal data.

Point 3. A figure of subjects wearing exoskeleton during data measurement must be added in Section 2.3;

Response 3: Thank you for this suggestion, a figure has been added in section 2.2.

Point 4. Section 2.4, line 135, "Baseline characteristics (age, ...", the right bracket is missing, which may confuse readers.

Response 4: The right bracket has now been added.

Point 5. There is no enough description and summarize on results, and the results description is messy. For example, Section 3.4, line 175, the author should emphasize the key points of relevant results instead of listing like a experiment report;

Response 5: We understand the reviewer but this study needed a very comprehensive description of both patients’ characteristics (demographic, clinical and kinematics) and treatment provided to patients over the rehabilitation program; so, it implied lots of descriptive statistics to present. However, we only showed significant differences from statistical analysis in the result section.

Point 6. There are many abbreviations in Abstract but seem not be used in text part;

Response 6: Abbreviations have now been removed from the abstract.

Point 7. The innovation and contribution of this article is not clear.

Response 7: the description of dose in non-pharmaceutical treatments is a challenge for clinicians and researchers in neurorehabilitation field. Lack of information and understanding of dose is probably a key reason that limits understanding of the effects of physical interventions and which populations / subgroups should be specifically targeted by interventions.

In this study, we provide a comprehensive description of the different dimensions of dose in a post stroke rehabilitation program as robotic devices offer a suitable paradigm to address this issue (automatic recording of a large amount of data and various physical modalities).

We believe that this study can be an interesting contribution on this topic in line with the works of The Stroke Recovery and Rehabilitation Roundtable Taskforce (Bernhardt J, Hayward KS…).

Reviewer 3 Report

The topic is well promising and scientifically sound. Authors must make the effort to take care of the form issues, such as FMA must declare where the first time abbreviation is used, among others. In addition, the contribution must be detailed in the conclusion section, including the modalities (assisted, unassisted). In the conclusion section, highlight: errors, correlation, convergence, testing validation, Standandard and guidelines (ISO, ASTM, NIST, RESNA, FDA, etc) applicable to Performance-based robotic training, application of findings; and future works.

Author Response

Response to Reviewer 3 Comments

Point 1: The topic is well promising and scientifically sound. Authors must make the effort to take care of the form issues, such as FMA must declare where the first time abbreviation is used, among others.

Response 1: Thank you for pointing this out, abbreviations have been verified.

Point 2: In addition, the contribution must be detailed in the conclusion section, including the modalities (assisted, unassisted).

Response 2: The contribution has been added in conclusion section.

Point 3: In the conclusion section, highlight: errors, correlation, convergence, testing validation, Standandard and guidelines (ISO, ASTM, NIST, RESNA, FDA, etc) applicable to Performance-based robotic training, application of findings; and future works

Response 3: A sentence for engineers and researchers has been added at the end of the conclusion.

Round 2

Reviewer 2 Report

Accept.